# Spatio-Temporal Patterns of Dengue Incidence in Medan City, North Sumatera, Indonesia

**DOI:** 10.3390/tropicalmed6010030

**Published:** 2021-03-05

**Authors:** Ayodhia Pitaloka Pasaribu, Tsheten Tsheten, Muhammad Yamin, Yulia Maryani, Fahmi Fahmi, Archie C. A. Clements, Darren J. Gray, Kinley Wangdi

**Affiliations:** 1Department of Pediatrics, Medical School, Universitas Sumatera Utara, Medan 20155, North Sumatera, Indonesia; 2Department of Global Health, Research School of Population Health, The Australian National University, Acton, Canberra, ACT 2601, Australia; tsheten.tsheten@anu.edu.au (T.T.); darren.gray@anu.edu.au (D.J.G.); kinley.wangdi@anu.edu.au (K.W.); 3Medical School, Universitas Sumatera Utara, Medan 20155, North Sumatera, Indonesia; yamien.m@gmail.com; 4North Sumatera Provincial Health Office, Medan 20232, North Sumatera, Indonesia; yuliamaryani.dr@gmail.com; 5Faculty of Engineering, Universitas Sumatera Utara, Medan 20155, North Sumatera, Indonesia; fahmimn@usu.ac.id; 6Faculty of Health Sciences, Curtin University, Perth, WA 6102, Australia; archie.clements@curtin.edu.au; 7Telethon Kids Institute, Nedlands, WA 6009, Australia

**Keywords:** dengue, spatio-temporal, clustering, Medan, Indonesia

## Abstract

Dengue has been a perennial public health problem in Medan city, North Sumatera, despite the widespread implementation of dengue control. Understanding the spatial and temporal pattern of dengue is critical for effective implementation of dengue control strategies. This study aimed to characterize the epidemiology and spatio-temporal patterns of dengue in Medan City, Indonesia. Data on dengue incidence were obtained from January 2016 to December 2019. Kulldorff’s space-time scan statistic was used to identify dengue clusters. The Getis-Ord Gi* and Anselin Local Moran’s I statistics were used for further characterisation of dengue hotspots and cold spots. Results: A total of 5556 cases were reported from 151 villages across 21 districts in Medan City. Annual incidence in villages varied from zero to 439.32 per 100,000 inhabitants. According to Kulldorf’s space-time scan statistic, the most likely cluster was located in 27 villages in the south-west of Medan between January 2016 and February 2017, with a relative risk (RR) of 2.47. Getis-Ord Gi^*^ and LISA statistics also identified these villages as hotpot areas. Significant space-time dengue clusters were identified during the study period. These clusters could be prioritized for resource allocation for more efficient prevention and control of dengue.

## 1. Introduction

Dengue is among the most important vector-borne diseases (VBDs) in terms of impact on the health of populations globally. Dengue fever (DF), together with the associated dengue haemorrhagic fever (DHF), has been the world’s fastest-growing VBD over recent decades, with a 30-fold increase [1]. Today, transmission occurs in 128 countries, 100 of which are endemic to DF [2]. In 2013, 390 million dengue infections were reported, of which 96 million manifested into clinical disease with associated DHF and dengue shock syndrome (DSS). DF is reported in all countries of the World Health Organization (WHO) South-East Asia Region (SEAR), except North Korea, and this region accounts for up to two-thirds of the global DF burden [3]. It is estimated that more than 4 million cases have occurred worldwide in the past 12 years [4]. It is projected that by 2085, more than 50% of the global population will be threatened by DF [5], and the densely populated SEAR is likely to be severely affected [6].

DF is caused by four serotypes of the dengue virus (DENV 1, 2, 3, and 4) that belong to the *Flaviviridae* family. DENV is transmitted by *Aedes aegypti* and *Ae. albopictus* mosquitoes [5,7,8]. DENV 2 and 3 are more commonly associated with hospitalization and greater mortality than DENV 1 and DENV 4 [9,10].

Historically, the first cases of DF in Indonesia were reported in 1968 from the cities of Jakarta and Surabaya. However, DF is now reported in all 34 provinces and 85% of districts [11], and all four serotypes are in circulation [12]. The transmission of DF is fuelled by increased population density, increased mobility of people, and inadequate sanitation. The country experiences year-round transmission of dengue, and bears the highest DF-related disease burden in the SEAR [13]. From 2004–2010, Indonesia reported the second-highest number of cases after Brazil worldwide [14]. In 2016, DF incidence was 78.9 cases per 100,000 population, with a case fatality rate of 0.8% [15]. Dengue is currently recognized as the most common cause of hospitalization among acute febrile illness cases in Indonesia. By the age of 12 years, more than 90% of the children have already seroconverted due to dengue infection [16]. This suggests the need to institute better epidemiological surveillance and control strategies for dengue in the country.

Studies in the past have utilized spatial and temporal epidemiological tools (such as models and maps) to support local health authorities to implement timely and targeted public health interventions and make better planning for dengue control [17,18,19,20]. These studies have unravelled high-risk areas and a clustered pattern of dengue incidence. Reliable information on areas at risk of dengue is essential to understand variations in local disease epidemiology and plan appropriate control strategies. The study presented here aims to characterize dengue risk in Medan city, North Sumatera, Indonesia using spatial and temporal analysis, and to evaluate clusters in terms of their location, timing, and duration.

## 2. Materials and Methods

### 2.1. Study Design

A descriptive study was conducted using routinely collected surveillance data for dengue. First, we evaluated the incidence and seasonality of dengue over the study period from 2016–2019. Secondly, space-time cluster analysis was performed to use Kulldorff’s spatial scan stastistic to evaluate and identify areas at risk for dengue to target prevention and control strategies. These areas at risk for dengue were further assessed by hotspot analysis that quantifies the existence of spatial autocorrelation.

### 2.2. Study Setting

Indonesia is an archipelago country administratively divided into 34 provinces, 98 cities, and more than 415 regencies. Each of these administrative areas is further broken down into thousands of districts and villages.

This study was conducted in Medan city, the capital of North Sumatera, located between 3°27′ N and 3°47′ N latitude and 98°35′ E and 98°44′ E longitude. The city occupies the northern part of the province (Figure 1) and is administratively divided into 21 districts and 151 villages. The population in the city is 2,277,601, which was 15% of the entire population in North Sumatera province in 2019. Medan features a tropical climate with a dry season (April–October) and a wet season (November–March). The average temperature in the city ranges from 26.1–28.8 °C, while the monthly total rainfall varies from 19.8–509 mm [21].

Historically, dengue virus in Indonesia was first isolated in the 1970s in Medan, Jakarta, and Semarang [22]. During that time, DENV-2 was the predominant serotype. However, DENV-3 and DENV-4 were isolated from DHF patients in Medan in recent years [23]. The city has now become endemic for dengue, with a high incidence of dengue infection reported every year. In the past, the city has been harder hit by dengue than other regencies and cities in North Sumatera. In 2017, Medan City had 1214 cases and 11 deaths, the highest numbers in the province [24].

### 2.3. Data Source

Dengue is a notifiable disease, and reporting by both the public and private health sectors is mandated by law in Indonesia [25]. According to the Ministry of Health of Indonesia, a reportable dengue case is defined by an acute fever with a positive serological test for dengue (either NS1 or IgM and Ig G) or any increase in hematocrit and decrease in platelet count [26].

The Medan dengue surveillance database was the source of the dengue incidence data used for this study. The computerized dataset included monthly numbers of dengue cases and deaths collected from the Medan City Health Office between January 2016 and December 2019. Population data were obtained from the Central Bureau of Statistics, and a shapefile for the administrative sub-divisions (villages) of Medan City was obtained from an online source [27]. Latitudes and longitudes of the centroids of each village polygon were used as the coordinates for spatial analysis.

### 2.4. Geographical Distribution of Dengue Incidence

The annual cumulative dengue incidence was calculated for each village by dividing the total number of annual cases in each village by the corresponding village population multiplied by 100,000. The annual incidence was then represented in a panel of choropleth maps, created using the geographical information system ArcGIS version 10.7.1 (ESRI, Redlands, CA, USA).

### 2.5. Temporal Trends of Dengue

Time-series seasonal decomposition analysis was performed to explore the seasonality of dengue. The monthly mean number of dengue cases was calculated for the full-time series (January 2016–December 2019). This monthly time-series was then decomposed into three temporal components using locally-weighted regression or loess: seasonality, trend (i.e., annual average dengue incidence), and residual variability [28,29,30,31]. The function *stl* and parameter setting *periodic* in R studio was used to break down the time-series data and extract all its components [32].

### 2.6. Spatio-Temporal Analysis

A retrospective space-time analysis was performed to identify areas and periods with a significantly higher than average risk of dengue using Kulldorff’s space-time scan statistic, which has been widely used in epidemiological studies of dengue [18,33,34]. The space-time scan statistic was defined by a cylindrical window, the base of which is centred around one of all possible village centroids located throughout the study region, with varying radius (representing the spatial dimensions of the cluster) and height (representing the temporal dimensions of the cluster). Cases were assumed to be Poisson-distributed with constant risk over space and time under the null hypothesis.

The maximum spatial cluster size was set at 20% of the total population at risk and 50% of the study period, with data aggregated by village and month. A criterion of no geographical overlap was used to report secondary clusters.

In the first phase of the analysis, the window is moved in space and time to cover each possible geographic location and size, and each possible time period [35]. For each window, the log-likelihood ratio (LLR) was computed by counting the observed and expected number of cases inside and outside the significant window. In the second phase, a Monte Carlo simulation with 999 repetitions was used to test the null hypothesis of the same relative risk between the clusters. The cluster with the highest LLR was considered as the most likely cluster, and the remaining clusters were denoted as secondary clusters. A significance level of *p* < 0.05 was considered significant. The Kulldorff’s space-time scan statistic was implemented using SaTScan^TM^ version 9.6 (Martin Kulldorff and Information Management Services Inc., Cambridge, MA, USA) [36] and the clusters were mapped with ArcGIS.

### 2.7. Hotspot Analysis

The presence and nature of spatial autocorrelation that suggests dengue clustering was assessed by Anselin Local Moran’s I and Getis-Ord statistic (Gi*) [37,38,39,40,41]. Moran’s I tests the null hypothesis that observed values at one location are independent of observed values at other locations (i.e., dengue incidence is randomly distributed). Its values can range from −1 to 1, where positive values (observed Moran’s I value is larger than the expected value) indicate the presence of spatial clustering of similar values (i.e., either hot or cold), zero means total spatial randomness, and negative values (Moran’s expected value is smaller than the observed value) indicate dissimilar values clustered next to one another (i.e., locations with high values surrounded by neighbours with low values and vice versa) [38,42]. Moran’s I was used to classify study areas into hotspots (high–high cluster), cold spots (low–low cluster), and spatial outliers (high–low or low–high cluster).

In addition, the local Getis-Ord statistic (Gi*) was used to indicate the intensity and stability of hotspot/cold spot clusters [40,42]. The G* statistic works by comparing the local mean rate (i.e., the rates for a target location and its neighbourhood) to the global mean rate (the rates for all locations). The Gi* statistics return a Z-score and *p*-value for each location and indicate whether the local and global means are significantly different or not. Location with a statistically significant and larger Z-score will have a more intense cluster of high values (hotspot), where it is very unlikely that the spatial clustering of high values is the result of a random spatial process; and locations with a statistically significant and smaller Z-score will have more intense clustering of low values (cold spots) [40]. ArcGIS was used hotspot analysis.

### 2.8. Ethical Consideration

Ethical clearance was obtained from the Medical Faculty of Universitas Sumatera Utara Health Research Ethical Committee No:145/TGL/KEPK FK USU-RSUP HAM/2020.

## 3. Result

### 3.1. Dengue Incidence

A total of 5556 dengue cases were reported in Medan city in the four years from 2016–2019. More than half of the cases in 2016, 2018, and 2019 belonged to the age group >14 years. Males were predominantly affected by dengue in all four years (Table 1).

Villages that reported an incidence of >350 cases per 100,000 people included Anggrung (Medan Polonia district) and Simalinkar B (Medan Tuntungan district) in 2016, and Suka Maju (Medan Johor district) and Nama Gajah (Medan Tuntungan district) in 2018. In contrast, Sukaramai II (Medan area district), Sei Rengas I (Medan Kota district), Pasar Baru (Medan Kota district), Simpang Tangjung (Medan Sunggal district), and Gang Buntu (Medan Timur district) did not report any cases during the study period (Figure 2).

### 3.2. Temporal Trends and Seasonality

The cumulative incidence per 100,000 inhabitants was 81.09, 54.09, 65.82, and 46.89 in 2016, 2017, 2018, and 2019, respectively. During the wet season (November–May), the frequency of dengue cases nearly doubled as compared to the dry season in all years, and the seasonal decomposition plot demonstrated strong seasonality, with a peak in January/February in each year. Variation in the remainder component was lower than for the seasonal and interannual varation (Figure 3).

### 3.3. Space-Time Clusters

There were seven statistically significant high-risk space-time clusters of dengue, as illustrated in Figure 4. Four of these clusters had a radius > 3 km, incorporating multiple contiguous villages, while two villages (i.e., Suka Maju in Medan Johor district) and Petisah Tengah (Medan Petisah) were depicted as individual-village clusters. The largest cluster (i.e., most likely cluster) with a relative risk of 2.47 (*p* < 0.001) was located in the south-west of Medan in January 2016–February 2017. This cluster had 28 contiguous villages. The largest secondary cluster (radius: 3.75 km) was located in the south-east of Medan and had a relative risk of 2.09 (*p* < 0.0001). The cluster occurred between October 2018 and January 2019. Although the size of the cluster was smaller than the most likely cluster, it contained more villages (totalling 32) (Table 2). Villages in these clusters were located in districts of Medan Denai, Medan area, Medan Kota, Medan Timur, Medan Perjuangan, and Medan Tembung.

The next largest secondary cluster (radius: 3.64 km) was detected between December 2018 and February 2019 in north-east Medan, and consisted of five villages across two districts, namely, Medan Deli and Medan Labuhan. The remaining clusters included two villages in the Medan Balawan district in the north-west (radius: 3.05 km), three villages in the Medan Maimun district in the south-west (radius: 0.57 km), and one village in each of the Medan Johor (radius: 0 km) and Medan Petisah districts in the south-west (radius: 0 km).

### 3.4. Hotspot and Coldspots

The Anselin Local Moran’s I showed statistically significant hotspot clustering of dengue in eight villages in 2016 in the south-west, eight villages in 2017 in the south-west, south-east, and north-east, nine in 2018 in the south-west and north-east, and 12 in 2019 in the south-west and north-east of Medan (Figure 5). Applying Getis G* statistic hotspot analysis to dengue incidence to each village revealed statistically significant (*p* < 0.01) hotspots for 10 villages in 2016 in the south-west, seven in 2017 in the south-west and north-east, seven in 2018 in the south-west, and 14 in 2019 in the south-west and north-east of Medan (Figure 6). A large cold-spot was consistently found using both analytical tecniques for all four years in the centre of the study area. Overall, the analyses consistently identified hotspots in the same areas.

## 4. Discussion

Temporal trend and seasonality analysis revealed fluctuations in annual dengue incidence with peaks in incidence taking place at the beginning of every year. Multiple analytical techniques showed consistent, significant high-risk spatoptemporal clusters of dengue in Medan, with the primary cluster identified by Kulldorff’s space-time scan statistic being in the southwest of the City. Therefore, there is scope for applying pragmatic public health interventions for dengue at fine spatial scale, to subdivisions of large metropolitan areas, such as villiges within Medan city.

The study revealed an almost even distribution of dengue incidence between children and adults, which is consistent with reports from the 2017 dengue outbreak in Vietnam [43] and Timor-Leste [30]. More cases were identified in males than females, which was also the case in other studies in Indonesia [16] and Nepal [44]. The higher risk of infection among males could be related to greater exposure during outdoor occupations and leisure activities. However, other studies reported a higher incidence of dengue in women [30,45].

The interannual fluctuation of dengue incidence represents a typical dengue epidemic cycle with the intense transmission of dengue every 3–5 years [46]. This interannual periodicity occurs because infection with dengue provides substantial transient and short-term protection against other serotypes for approximately 1–3 years, so subsequent large epidemics do not occur during this time [47]. The population becomes susceptible to future outbreak/s with the waning of this transient immunity and low levels of herd immunity, which is intensified by implementation of vector control activities following outbreaks that protect people from *Aedes* mosquito bites and dampen levels of acquiring natural immunity [48].

Dengue was highly seasonal in Medan City, with a higher incidence in the peak rainy season. This finding is consistent with studies reported from other parts of Indonesia [17,20,49]. Increased rainfall facilitates the growth of vector populations by providing water for aquatic breeding sites, highlighting the important role of the climate in the transmission of dengue.

Our space-time model yielded seven different clusters in different time-periods. These clusters were widely spread across different locations in Medan City, and villages concentrated in these clusters have a statistically significantly higher number of observed cases than the expected number of cases. High-risk villages, as indicated by the most likely cluster, were mostly concentrated in the south-west of Medan, while other clusters were more spread out. These areas are characterised by a crowded environment, increased human activity, poor housing infrastructure, poor knowledge on dengue mosquitoes, low income levels, and poor hygiene and sanitation [50,51]. Due to scarcity of water, inhabitants living in these areas have to store water in water storage containers, which provide suitable breeding habitats for *Aedes* mosquitoes [52], further exacerbating dengue risk.

Hotspots identified using Local Moran’s I and Getis G* statistics were consistent with those identified by Kulldorff’s space-time scan statistic. Clustering of villages with higher dengue incidence surrounded by villages having higher dengue incidence (HH) or hotspots was predominantly observed in the south-west of Medan throughout the study period. Similar findings were also reported in other studies [53,54]. The consistency of these findings adds confidence in the existence of hotspots in particular locations, strengthening the argument that spatially targeted resource allocation is likely to improve the cost for effectiveness of dengue control programmes.

The most important limitation of the current study is the potential for inconsistent reporting of dengue across health centres, and the likelihood that not all cases would have been captured by the passive surveillance system. However, we do not have any evidence to suggest an underlying pattern or under-reporting, particularly within a relatively small and highly urbanized geographical area. The current study does, however, demonstrate how routinely collected information can be utilised to gain additional insight into disease patterns using spatiotemporal analytical approaches.

## 5. Conclusions

Dengue cases were clustered in space and time within a large metropolitan city in Indonesia, as identified consistently using multiple analytical methods. High-risk metropolitan sub-divisions (in this case, villages) can be targeted using focused strategies to faciltiate a more cost-effective approach to dengue control and prevention.

## Figures and Tables

**Figure 1 tropicalmed-06-00030-f001:**
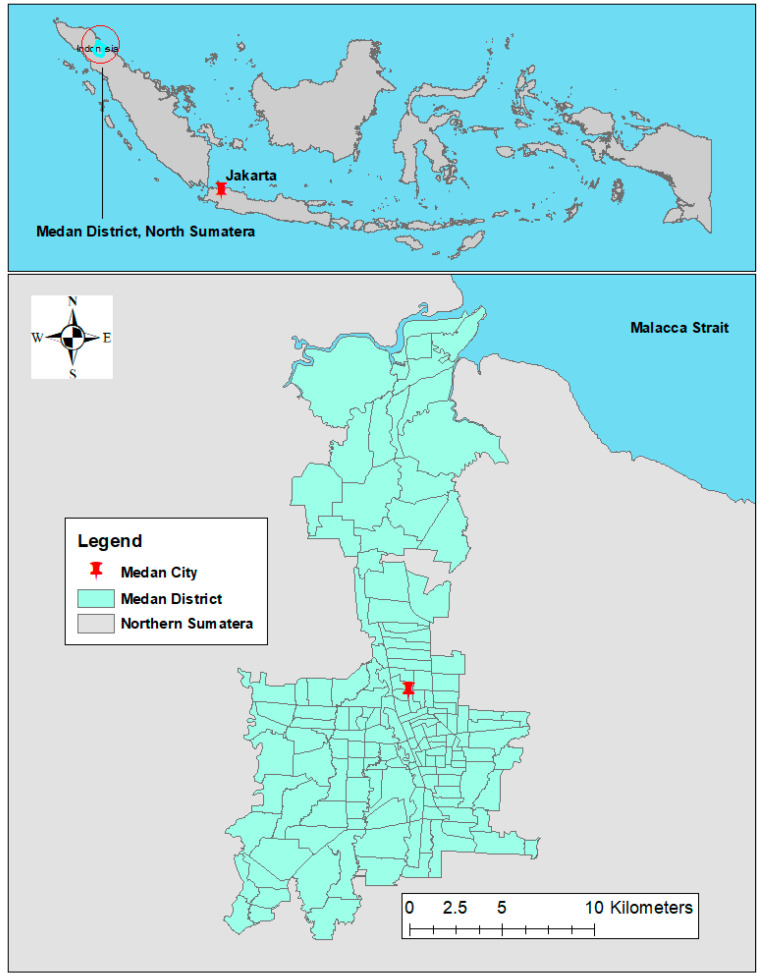
Administrative boundary of 151 villages in Medan City, North Sumatera, Indonesia.

**Figure 2 tropicalmed-06-00030-f002:**
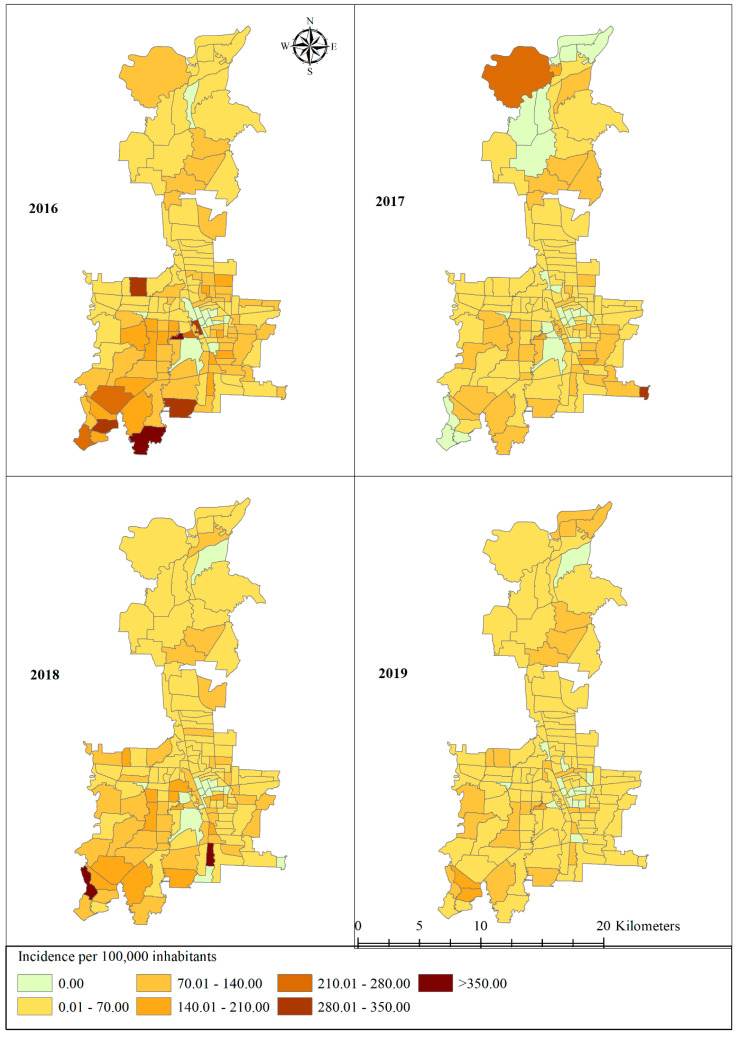
Distribution of dengue incidence at the village level in Medan city, North Sumatera, Indonesia, 2016–2019.

**Figure 3 tropicalmed-06-00030-f003:**
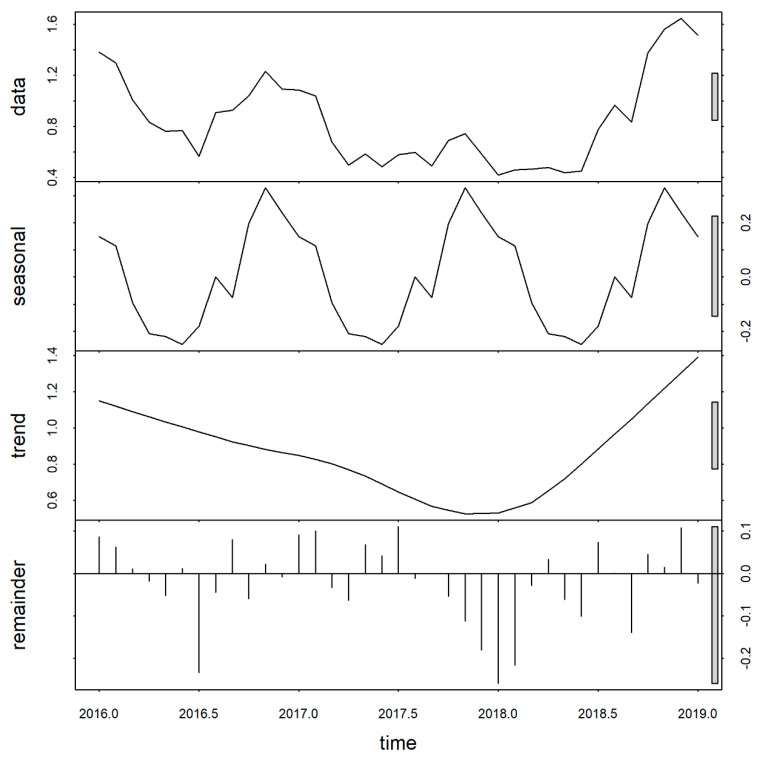
Temporal decomposition of dengue incidence in Medan city, North Sumatera, Indonesia, 2016–2019.

**Figure 4 tropicalmed-06-00030-f004:**
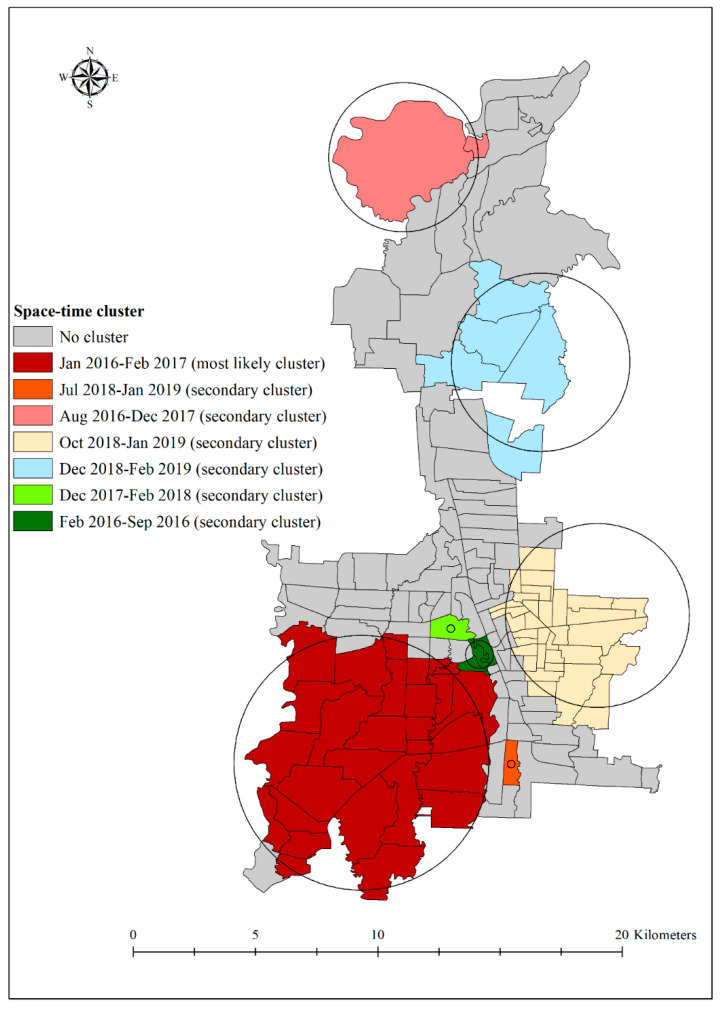
A significantly high rate of spatio-temporal clusters using retrospective space-time analysis of dengue incidence in Medan city, North Sumatera, Indonesia, 2016–2019.

**Figure 5 tropicalmed-06-00030-f005:**
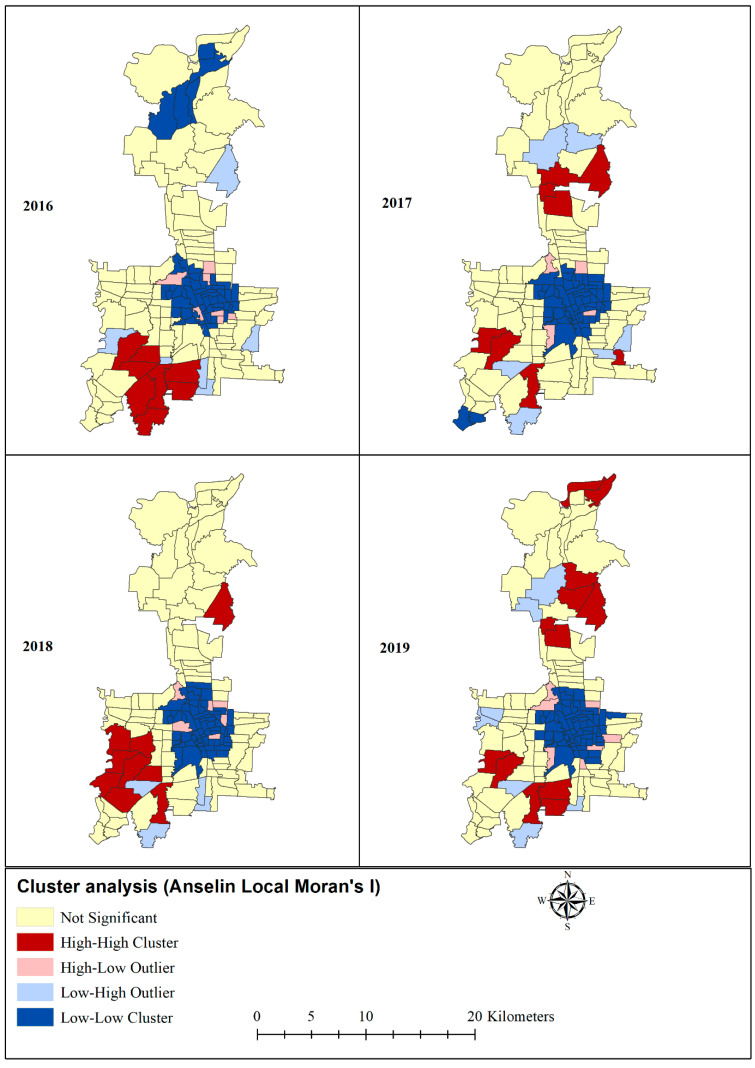
Dengue incidence clusters and outliers by village based on Anselin Local Moran’s I statistic in Medan city, North Sumatera, Indonesia.

**Figure 6 tropicalmed-06-00030-f006:**
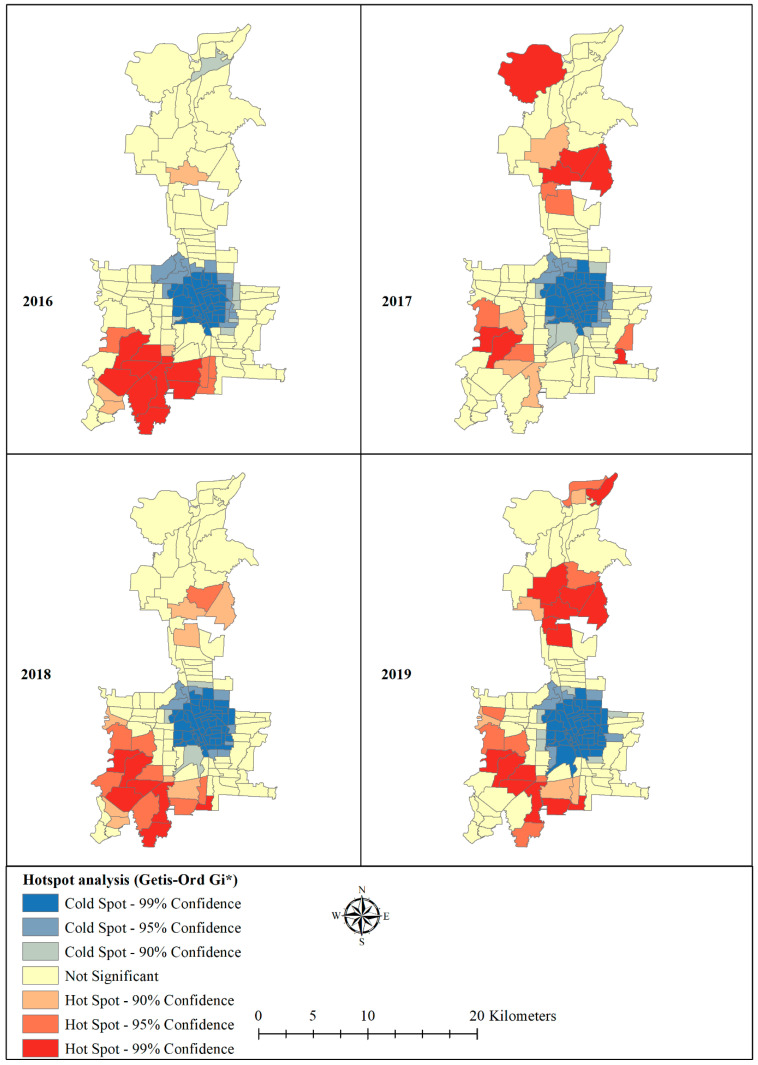
Statistically significant hotspots and cold spot locations of dengue incidence in Medan city, North Sumatera, Indonesia.

**Table 1 tropicalmed-06-00030-t001:** Demographic and seasonal variation in numbers of dengue cases in Medan city, North Sumatera, Indonesia, 2016–2019.

Parameters		2016		2017		2018		2019	
Age (years)		n	%	n	%	n	%	n	%
	Under-14	840	47.09	623	51.32	726	48.72	482	45.13
	Above-14	944	52.91	591	48.68	764	51.28	586	54.87
Sex									
	Male	947	53.08	631	51.98	769	51.61	579	54.21
	Female	837	46.92	583	48.02	721	48.39	489	45.79
Season									
	Wet (Nov–May)	1,149	64.41	786	64.74	825	55.37	700	65.54
	Dry (Jun–Oct)	635	35.59	428	35.26	665	44.63	368	34.46
Total		1784	100.00	1214	100.00	1490	100.00	1068	100.00

**Table 2 tropicalmed-06-00030-t002:** Significant space-time clusters and associated characteristics of dengue incidence in Medan city, North Sumatera, 2016–2019.

Cluster Period	Villages (Number)	Radius (km)	Observed Cases	Expected Cases	RR	LLR	*p* Value
Jan 2016–Feb 2017	28	5.2	716	314	2.47	203.484	<0.0001
Jul 2018–Jan 2019	1	0	44	4	11.53	67.232	<0.0001
Aug 2016–Dec 2017	2	3.05	89	24	3.71	51.229	<0.0001
Oct 2018–Jan 2019	32	3.75	187	91	2.09	39.475	<0.0001
Dec 2018–Feb 2019	5	3.64	65	21	3.1	29.282	<0.0001
Dec 2017–Feb 2018	1	0	14	2	9.62	19.135	<0.0001
Feb 2016–Sep 2016	3	0.57	24	5	4.92	19.065	<0.0001

Note: RR: Relative Risk; LLR: Log Likelihood Ratio.

## Data Availability

Data used for our study is available in the Appendix A.

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
