# Peer review of "Spatio-Temporal Patterns of Dengue Incidence in Medan City, North Sumatera, Indonesia"

_tropicalmed, 2021, doi:10.3390/tropicalmed6010030_

Round 1

Reviewer 1 Report

This is a relatively technical paper demonstrating spatio-temporal variations in dengue incidents in Medan city

I was a little unsure of the spelling of Sumatra vs Sumatera - I gather the former is the English spelling and the latter is the spelling in Indonesian. I would defer to your judgement over that.

References nine and 10 may be applicable to the country settings for those papers and the increase hospitalisation and mortality associated with dengue 2 and 3 may not be generalisable to other countries.

Under the section relating to data source (before reference 26) the criteria for dengue appears to be quite broad - there are a lot of diseases that cause an increase in haematocrit and a decrease in platelet count and I'm sure these are not counted as dengue cases. The NS1 is quite specific, the IgM is less specific and the IgG could indicate infection any time in the past so as read, patients with IgM and IgG would represent patients who are quite late in their primary illness or having a secondary infection. Many patients in the early stage of infection will have only IgM- and I would be surprised if these were excluded. Could you please just make a little clearer the definitions for dengue as described in the reference.  In the reference list reference 26 does not have enough information available to find the publication - is there a URL?

For figure 3 - is data for 2019 missing? - The x-axis is a little unclear and I didn't understand the third box labelled "trend" 

As a clinician I was not able to understand the statistics used in a found the terms "most likely cluster",  "primary cluster" and "secondary cluster" confusing - could you included plain English explanation for these terms.

Author Response

  1. I was a little unsure of the spelling of Sumatra vs Sumatera - I gather the former is the English spelling and the latter is the spelling in Indonesian. I would defer to your judgement over that.

Response: Thanks for highlighting two spellings for Sumatera and the correct spelling is North Sumatera as outlined in the title.

  1. References nine and 10 may be applicable to the country settings for those papers and the increase hospitalisation and mortality associated with dengue 2 and 3 may not be generalisable to other countries.

Response: We changed ref 9 and 10 to references from Indonesia that also mentioned DEN-3 as severe form of dengue.

  1. Under the section relating to data source (before reference 26) the criteria for dengue appears to be quite broad - there are a lot of diseases that cause an increase in haematocrit and a decrease in platelet count and I'm sure these are not counted as dengue cases. The NS1 is quite specific, the IgM is less specific and the IgG could indicate infection any time in the past so as read, patients with IgM and IgG would represent patients who are quite late in their primary illness or having a secondary infection. Many patients in the early stage of infection will have only IgM- and I would be surprised if these were excluded. Could you please just make a little clearer the definitions for dengue as described in the reference.  In the reference list reference 26 does not have enough information available to find the publication - is there a URL?

Response: We have added the following amendments in the revised manuscript (page 5).

The confirmed dengue from the National guideline is based on clinical manifestation and lab could be NS1, or IgM for dengue or IgM and IgG for dengue hemorrhagic fever. We include anyone with dengue infection in our study, could be dengue fever (DF) or DHF. The guideline is in hardcopy version.

  1. For figure 3 - is data for 2019 missing? - The x-axis is a little unclear and I didn't understand the third box labelled "trend" 

Response: Figure 3 is updated with the data included for the study. Trend here refers to annual average dengue incidence as updated in the manuscript in page 4, sub-section “Temporal trends of dengue”. The x-axis in figure 3 is common for all the four components, and includes both year and month.

  1. As a clinician I was not able to understand the statistics used in a found the terms "most likely cluster",  "primary cluster" and "secondary cluster" confusing - could you included plain English explanation for these terms.

Response: The definitions of “most likely cluster” and secondary cluster” has been added in the revised manuscript to make it clear (pages 4).

Reviewer 2 Report

Estimated Editors,

Estimated Authors,

I gratulate with Pasaribu et al. for this high-quality paper dealing with Spatio-temporal patterns of dengue incidence in Medan city, North Sumatera, Indonesia. Authors have analyzed data from incident cases of Dengue in the aforementioned city of Medan, referring to very accurate and detailed spatio-temporal analysis methods. 

In my opinion, no significant amendments or improvements are required, and the paper may be published as it is. 

Author Response

I congratulate with Pasaribu et al. for this high-quality paper dealing with Spatio-temporal patterns of dengue incidence in Medan city, North Sumatera, Indonesia. Authors have analyzed data from incident cases of Dengue in the aforementioned city of Medan, referring to very accurate and detailed spatio-temporal analysis methods. 

In my opinion, no significant amendments or improvements are required, and the paper may be published as it is. 

Response: Thanks for reviewing this paper.

Round 2

Reviewer 1 Report

authors have responded to queries satisfactorilly